# Whole Slide Image Classification of Gastric Cancer using Convolutional Neural Networks

**Junni Shou[1], Yan Li[1], Guanzhen Yu[2,\*], and Guannan Li[1,\*]**

[1]Awakens Intelligence Technology Co., Ltd, China
[2]Department of Oncology, Longhua Hospital Affiliated to Shanghai University
of Traditional Chinese Medicine, Shanghai, China
*qiaoshanqian@aliyun.com, *guannan.li@awkint.com

## Abstract

Gastric cancer is one of the main causes of cancer and cancer-related mortality worldwide, and the diagnosis based on histopathology images is a gold standard for gastric cancer detection. However, manual diagnosis is labor-intensive and low in inter-observer agreement. Computer-aided image analysis method were thus developed to alleviate the workload of pathologists and overcome the problem of subjectivity. Histopathology image analysis using deep learning has been proved to give more promising results than traditional methods on many whole slide image cancer detection tasks, including breast cancer detection and prostate cancer detection. In this paper, we further studied a whole slide image classification method using Convolutional Neural Networks (CNNs) on gastric cancer data. The method classify a whole slide image based on patch-sized classification results. Various experiments for patch-level classification using different existing CNN architectures were conducted. Experiment results show that the architecture gives the state-of-the-art result in natural image classification tasks can also give impressive results in histopathology image classification tasks.

## 1   Introduction

Gastric cancer is the second most common cancer in China[3] and the third leading cause of cancer death worldwide[16]. Diagnoses in histopathology images is essential for assessing the tumor response and prognosis of patients to different treatments[11, 2, 5, 20]. Nevertheless, the manual pathological diagnoses are time-consuming, often require tedious and laborious work. Also, manual diagnoses could be subjective and difficult to standardize, leading low level diagnostic concordance[15, 4]. Therefore, computer-aided histopathology image analysis methods are developed to assist pathologists to improve the efficiency, accuracy and consistency of cancer detection[6, 7].

Recent works show great success in applying deep learning for histopathology image analysis. Specifically, Convolutional Neural Networks (CNNs) are applied to analyze the complicated histopathology images. This technique allows an image analysis method to be designed without specific field-related knowledge, and the model would learn all the features from images itself. Spanhol et al.[19] used a simple CNN architecture, AlexNet[10], to classify hematoxylin and eosin (H&E) stained breast histopathology images into two classes, benign and malignant. Small scaled input patch were used in their work. They then combined the results from patch classification to give the local-region-level classification. Subsequently, Araújoo et al.[1] extended the classification problem from 2-class to 4-class, and also experimented larger scaled input patches. While these works focused their studies in patch-level and local-region-level classifications, Litjens et al.[12], Wang et al.[21] and Liu et al.[13] further improved the image analysis methods, giving a whole-slide-level classification prediction.

Submitted to 1st Conference on Medical Imaging with Deep Learning (MIDL 2018). Do not distribute.

Table 1: Details of annotations given for the gastric cancer datasets

| | Training/Validation | Testing |
|---|---|---|
| Slide-level labels (WSIs) | 150 cancer + 39 normal | 110 cancer + 70 normal |
| Pixel-level annotations | 1500 region images (from 150 cancer WSIs, 10 from each slide) | 5 cancer WSIs |

Authors in works mentioned above proposed their classification methods for either breast cancer data or prostate cancer data, and Sharma et al.[17] later applied deep learning methods to the gastric cancer data. They proposed an introductory CNN architecture and compared the performance of it with AlexNet[10] and several other traditional methods. 15 whole slide images (WSIs) were used for extracting patches for training, validation and testing (11 for cancer classification and 4 for necrosis detection), and accuracies of 69.90% for cancer classification and 81.44% for necrosis detection were achieved for patch-level classifications.

In our work, with a larger gastric cancer dataset introduced, we evaluated the feasibility of a whole-slide-image-level classification method for gastric cancer. Additionally, to see would the architecture with more complicated structures outperform AlexNet[10] for histopathology images, different existing CNN architectures were assessed in patch-level classifications. The effect of different patch scales were also experimented. Finally, we achieved an accuracy of 98.698% for patch-level classification and an accuracy of 97.728% for slide-level classification.

## 2 Dataset

The gastric cancer dataset consists of 369 WSIs, each from a distinct patient who underwent curative surgery at Changhai Hospital in Shanghai, China, from 2001 to 2005. Mean age of these patients was 59 years old. The slides in the dataset were stained with hematoxylin and eosin (H&E), and digitized by MAGSCANNER KF-PRO-120[1] at magnification of $20\times$. The use of these slides has been approved by the Changhai Hospital Institutional Review Board.

Annotations of the data are given by expert pathologists, and presented in two different forms, pixel-level delineation of cancerous regions on images and cancer/normal labels for each slide. 1500 cancer region images (acquired from 150 WSIs, each with 10 region images), each of size $2048 \times 2048$ pixels, and another 5 cancer WSIs are given with pixel-level annotations (the 5 WSIs are exhaustively annotated). The 1500 cancer region images were used for extracting positive patches used for training and validating the patch-level classifier, and the 5 WSIs were used for positive patch extraction for testing the trained classifier. The patch extraction strategies would be further explained in details in the following sections. Total number of 369 WSIs are given with slide-level labels, and are split into 189 (150 cancer slides and 39 normal slides) for training of the slide-level classifier and 180 (110 cancer slides and 70 normal ones) for testing. The normal slides were also used for the negative patch extraction. The first 39 normal slides were used for extracting negative patches for training and validation of the patch-level classifier, whereas 5 out of the 70 normal slides were used for testing. Details of the annotations are summarized in Table 1.

## 3 Methods

The classification method consists of four steps: (1) image preprocessing to extract the tissue region; (2) patch-level classification using CNN; (3) cancer likelihood map generated from the patch-level classification results; (4) slide-level classification based on the likelihood map. Details are explained in the following sections.

---

[1]http://www.kfbio.cn/productshow.php?cid=27&id=43

## 3.1 Image Preprocessing

Most of the WSI area is non-informative background. These area would lead to unnecessary computational costs. To save computational costs and increase efficiency, we did image preprocessing to extract the tissue regions from the slide first.

Common thresholding algorithms were used in [12, 21] to extract the tissue region. These thresholding algorithms differentiate foreground and background objects by setting a threshold intensity, and simply grouping pixels with intensity higher than the threshold and lower than the threshold separately[14]. These methods are capable of separating the foreground objects from the blank background regions, however, it is unable to remove regions of glasses, glues and dirt, which would have pixels with similar intensities to tissue regions but different from blank regions. These useless regions would remain together with the tissue regions, causing computational costs and unnecessarily complicating the cancer detection problem since patches containing different forms of glass textures would also be required as normal patches for the training of the patch-wise classifier. Therefore, a tissue extraction method based on differences between R/G/B color channels was used. The blank background is close to white, whereas the regions of glasses, glues and dirt are generally greyish or close to black. Pixels with color close to black or white, or greyish colors would have relatively uniform values for R/G/B channels. In other words, the difference between the highest channel value and the lowest channel value of the pixel with those colors would be smaller than a certain threshold. Thus, pixels with channel value difference greater than the threshold would be marked as tissue regions, while the remaining would be regarded as the non-informative background. A threshold value of 25 was empirically obtained and was used to get the binary mask. Noises and small holes were later removed by morphological operations. Figure 1 shows the results of applying the tissue extraction method.

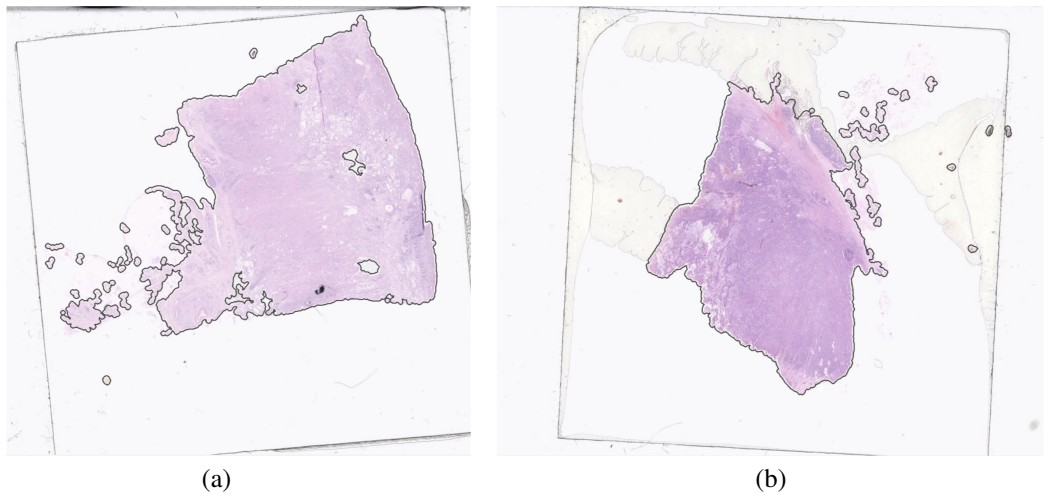

|     |     |
| --- | --- |
| (a) | (b) |

Figure 1: Examples show tissue regions extracted from WSI. The extracted tissue regions are successfully separated from background consists of blank regions, marks of glasses and glues, and dirt, and contoured with black curves.

## 3.2 Patch-wise Classification

The WSIs are large in size, hence it is impossible to directly put them as the input for the classification. One good way is to divide the tissue regions into small patches and the further slide-level classification could be done by combining all the results of the small patches. Since the objective is to give diagnostic results of cancer/normal for the slide, it is not important to precisely delineate the boundaries of cancerous regions on the slide. As a result, image classification models were considered rather than segmentation ones. For achieving better performance, CNN was used for patch-wise classifications. In the following, patch extraction strategies, data augmentation and detailed explanation of network architectures we used for training the patch-level classifier are presented.

### 3.2.1 Patch extraction

Patches for training, validation and testing sets were generated according to the pixel-level annotations given by pathologists. On images with pixel-level annotations, patches were extracted with a stride of 100 pixels in the tissue region and labeled as positive if the patch center located in the annotated cancerous region. Normal patches were generated randomly in the tissue regions of normal slides, and labeled as negative. To avoid bias to the patch dataset, the ratio of total amount of negative patches to total amount of positive patches was controlled to be roughly 1:1. Three patch sizes were extracted for further comparison: $120 \times 120$, $240 \times 240$ and $480 \times 480$.

### 3.2.2 Data augmentation

Data augmentation was utilized to obtain more robust models. To increase the size of the patch dataset, random cropping of sizes $112 \times 112$, $224 \times 224$ and $448 \times 448$ was applied to input patches of $120 \times 120$, $240 \times 240$ and $480 \times 480$ respectively during the training. Since features extracted from histopathology images should be orientation invariant, random flipping and rotation were also used. Vertical flipping and horizontal flipping would be applied to the input patches randomly with a probability of 0.5, and the patches would then be rotated by random multiples of $90°$. To combat the variations between different slides caused by, for example, different color staining, problem caused by overexpossure during scanning, etc., the brightness, contrast, saturation and hue of the patches were slightly adjusted by a random factor in each training epoch.

### 3.2.3 Using existing network architectures

We experimented with various previously existing CNN architectures to find the network architecture that suits the gastric cancer classification problem best. We started the evaluation with AlexNet[10], a network architecture simply composed of layers of convolution and pooling sequentially, followed by fully-connected layers. Next, VGG-16[18] was evaluated. VGG-16 has very similar "plain" network structures to AlexNet, but with more layers. We then experimented on more complex models, ResNets[8] and DenseNets[9]. These models have much deeper networks. ResNets[8] use identity-based shortcut connections to bypass the signal from previous layers to the next, alleviating degradation problems during the training for very deep networks. DenseNets[9] provide with a reformulation of the connection, which helps to train a deeper network but also substantially improves the parameter efficiency and better the generality of the trained model.

## 3.3 Cancer Likelihood Map

Tissue regions were extracted from the slide first, and patch-level inference was then carried out in a sliding window manner with strides of 28, 56, 112 and 224 pixels in the tissue regions. Smaller strides would lead to finer results but with more computation time. We did experiments to compare the results using different strides, and the experiment suggests that the results with stride of 56 was good enough to give visually smooth likelihood map and deliver enough information for later slide-level classification, while not being too time-consuming. Therefore, stride of 56 pixels was used for later experiments in this paper.

Classification results for small patches were then merged into the cancer likelihood map. Pixels in the patches predicted to be positive would be added by one on the map while the ones predicted to be negative would be remained as the original value. The final cancer likelihood map were then normalized to values in range [0,1] by dividing by

$$factor = \left(\frac{patch\ size}{stride}\right)^2 \tag{1}$$

## 3.4 Slide-level Classification Based on Cancer Likelihood Maps

After getting the cancer likelihood map for the slide, $N_t$ binary masks according to $N_t$ different thresholds of likelihood were obtained from the map, where $N_t$ is the number of thresholds used. For each threosholded binary mask, we collect 9 features, including area, solidity, eccentricity and extent of the largest component and the second largest component of the cancer area, and ratio between the total cancer area and the tissue area. Thus each slide could get $N_t \times 9$ features.

Table 2: Details of datasets for training the patch-level classifier

|  | Training | Validation | Testing |
| --- | --- | --- | --- |
| positive/negative patches | 135k/135k | 34k/34k | 50k/50k |
| Total | 270k | 68k | 100k |

189 slides (150 cancer slides and 39 normal slides) provided with slide-level labels were used for training the slide-level classifier. Cancer likelihood maps were first generated for these slides. With the features extracted from the map as the inputs and the slide-level label as the output label, a random forest classifier was trained to determine whether the slide should be predicted as cancer or normal. The performance of the slide-level classifier was further tested by the testing set consisting of 110 cancer slides and 70 normal slides.

## 4  Experiments

### 4.1  Patch-level Classification

1500 annotated $2048 \times 2048$ region images were used to generate a total amount of 169,275 positive patches for training and validation. Negative patches for training and validation were then extracted from 39 normal slides. In order to keep the ratio of the total amount of positives to negatives to be roughly 1:1, each normal slide was used for generating 4340 negative patches randomly positioned in the tissue regions of the normal slide. Accordingly, total amount of 338k positive and negative patches were generated. Then, these patches are equally divided into 5 groups. One of the groups was used as validation set whereas the remaining was used as the training set, leading to a training set of 270k patches and a validation set of 68k patches. Then, 5 annotated cancer slides and 5 normal slides from the additional 180 slides were used to produce the testing set. For each slide, 10k patches were extracted, making a testing set of 100k patches in total, consisting of 50k positives and 50k negatives. Details for each dataset are summarized in Table 2.

Because of the tremedous differences between histopathology images and natural images, we did not use any pretrained models and all networks were trained from scratch. The network was trained for 20 epochs, and the one with the highest accuracy on validation set was saved. Then the best model from last 20-epoch-training would be trained for another 20 epochs with a learning rate ten times smaller than before. Repeating for three times and the final network was obtained. All networks for comparison were acquired in the same way.

Table 3 shows the accuracy of different network architectures. Results in Table 3 indicate that the accuracy of a testing set is always lower than that of a validation set. This is because that the validation set was draw out from the same dataset where the training set was from, which means these two sets have though slightly different but similar patches. As for the testing set, patches were extracted from another 10 slides that had never been seen in the training process. Hence, patches in the testing set should be more different from the training patches, and that difference leads to the decrease in the accuracy. However, although there is a slight drop, the accuracy on the testing set is still very high. This may be due to the large amount of patches we used for training, and proper extensive data augmentation encourages the generality of the model and avoids over-fitting problems.

We first evaluated the performance of different network architectures with the same patch scale, $224 \times 224$. Although VGG-16[18] has a very simple and straightforward architecture, it still achieved surprisingly good result. This may be due to large amount of parameters in the VGG-16 architecture. Still, the highest accuracy was achieved by DenseNet-201[9]. DenseNet-201 has much fewer parameters than VGG-16, but its structure utilize features in an efficient way to avoid feature redundancy and help to generate a more compact network delivering better results. Regarding the fact that DenseNet-201 gave the best results for both validation and testing sets, we performed the following experiment using DenseNet-201.

Next, we compared the performance of DenseNet-201[9] using different input scales. Input size of $112 \times 112$ pixels gave much lower accuracy as expected, whereas model with $448 \times 448$ sized inputs gave sightly better results than $224 \times 224$. Since the improvement of model using $448 \times 448$ sized inputs was not very significant, about 0.1% in accuracy on training set and 0.2% on testing set, and it

Table 3: Patch-wise classification accuracy (%)

| Network | Input patch size | Validation | Testing |
|---|---|---|---|
| AlexNet | $224 \times 224$ | 98.156 | 96.722 |
| VGG-16 | $224 \times 224$ | 99.565 | 98.413 |
| ResNet-101 | $224 \times 224$ | 98.879 | 98.353 |
| ResNet-152 | $224 \times 224$ | 99.290 | 97.244 |
| DenseNet-121 | $224 \times 224$ | 98.444 | 98.153 |
| DenseNet-201 | $112 \times 112$ | 97.845 | 96.821 |
|  | $224 \times 224$ | **99.655** | **98.698** |
|  | $448 \times 448$ | **99.758** | **98.973** |

Table 4: Slide-level classification accuracy (%)

| Random forest classifier | Training | Testing |
|---|---|---|
| accuracy | 100.000 | 97.728 |
| sensitivity | 100.000 | 95.454 |
| specificity | 100.000 | 100.000 |

would cause substantial increase in computational costs. Considering the time constraint, we chose DenseNet-201 with $224 \times 224$ as the input size to finish the following experiments.

## 4.2 Slide-level Results

Once the trained network is obtained, it can be applied to the tissue region of the slide in a sliding window manner. The cancer likelihood map can be generated afterwards. An example[2,3] of cancer likelihood map for gastric cancer detection is shown in Figure 2(c). Figure 2(b) presents the corresponding ground truth annotation given by pathologists. Regions predicted with high likelihood of being cancerous are shown in red or yellow, whereas regions with low likelihood of cancer are shown in green or blue. Transparent areas indicate normal tissue regions. Most of the cancerous regions are correctly detected. Few false positives exist.

After getting the likelihood map, features were extracted from the map and fed as the training inputs to a Random Forest classifier. The slide-level label was used as the training ground truth. Dataset of 189 labeled slides (comprised of 150 cancer slides and 39 normal slides) was used as the training set and additional 180 slides (comprised of 110 cancer slides and 70 normal ones) were used as the testing set. We evaluated the performance of slide-level classifier with accuracy, sensitivity and specificity. Results for the slide-level classifier are summarized in Table 4. It can be seen from Table 4 that the classifier classified all normal slides in the testing set correctly but mis-classified several cancer slides. Most of the mis-classified cancer slides contains very few amount of cancerous area, like 0.23% of the tissue area. In the training set, the cancer slides contain an average cancerous region of 8.77% of the tissue regions (the least amount is 1.32% of the tissue region). Hence, even though the patch-level classifier is able to detect the cancerous regions in the slide, the post-processing slide-level classifier was trained to "assume" those detected cancerous regions to be false positives and gave incorrect slide-level classification results. In spite of this, the slide-level classifier is still able to give 100% of accuracy for gastric cancer detection if the slide contains cancerous area more than 1.5% of the tissue region.

---

[2]Full sized example image and the corresponding cancer likelihood map can be viewed by link: http://box.histogram.cn/s/i7Aune. The link is generated by HISTOGRAM™for data sharing. Likelihood map can be viewed by clicking the "heatmap" icon, and the pixel-level annotation can be viewed by clicking the eye icon.

[3]More full sized examples and corresponding maps can be viewed here: http://box.histogram.cn/s/OP0eNa

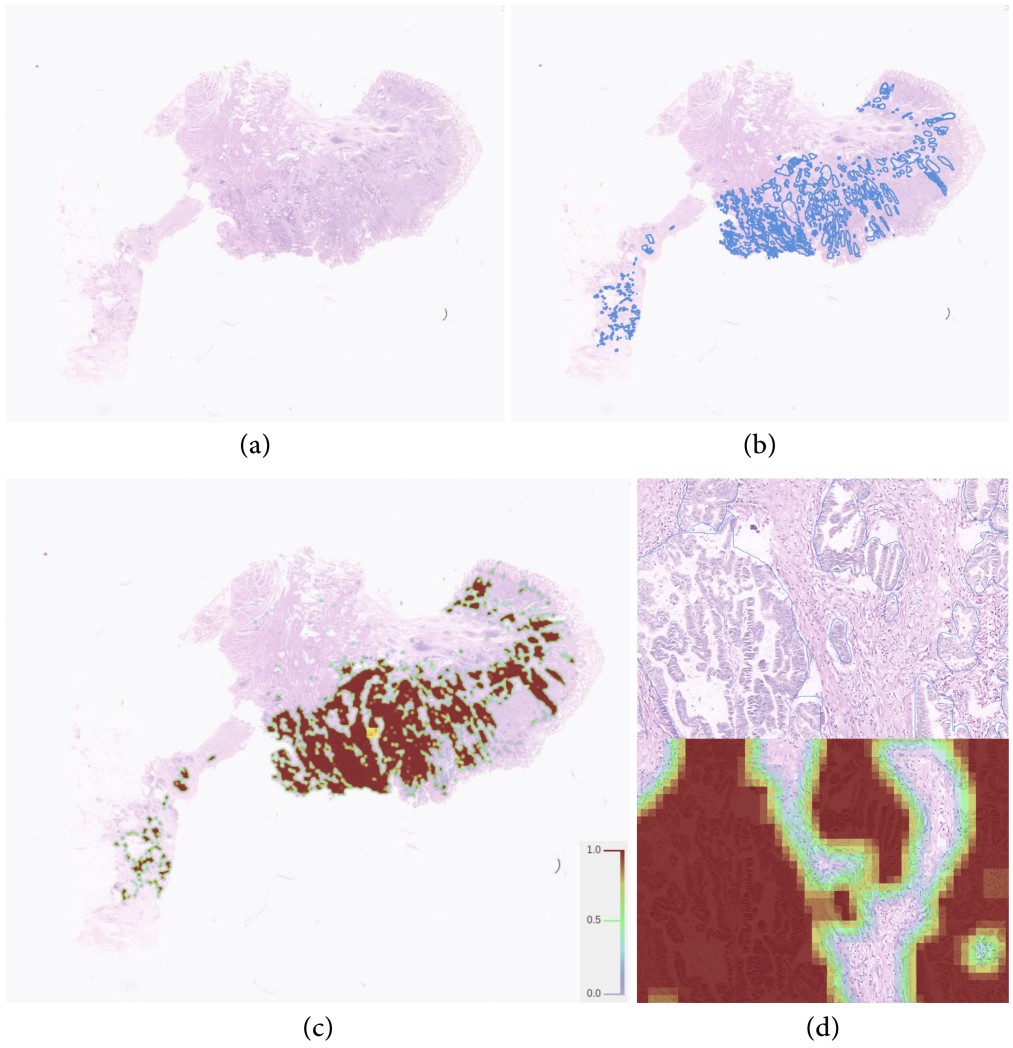

Figure 2: (a) Whole slide image of a tissue sample of gastric cancer. (b) The ground truth annotation of cancerous regions given by expert pathologists. (c) The predicted cancer likelihood map of the slide. (d) Zoom in on area indicated by the yellow square on (c) to a higher magnification ($10\times$). The top image is the annotation of the cancerous region, and the bottom image is the cancer likelihood map.

## 5   Conclusions

In this paper, the feasibility of a whole slide image classification method of gastric cancer using CNN is studied. An image preprocessing method is introduced to extract tissue regions from non-informative background, including blank regions, marks of glassess, glues and dirt. The whole slide classification is acquired by combining patch-level classification results. Patch extraction strategies are shown, and data augmentation is applied to increase the size of the training dataset for the patch-level classifier. Different existing CNN architectures are evaluated for patch-wise classification, and DenseNet-201 is reported to be the best network architecture for histopathology image classification of gastric cancer, giving an accuracy of 98.698% for the testing set. This leads to the conclusion that the best-in-class network architecture for natural image classification tasks can also give promising results in gastric cancer histopathology image classification. The cancer likelihood map for whole slide image of gastric cancer is produced by aggregating the patch-wise classification results. Final slide-level classifier is trained based on Random Forest classifier, using features extracted from the

corresponding cancer likelihood map as the inputs. Experiment demonstrates that the slide-level classifier achieves an accuracy of 97.728% for the testing set. We thus conclude that the whole slide image classification method is useful for gastric cancer detection.

Future work can extend the binary classification of cancer/normal to multi-class classification, to distinguish between various sub-types of the gastric cancer. Moreover, data used for this work are from the same center using the same scanner. Further studies can explore the method with data from multiple centers and different digitization equipments.

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
