# OpenReview forum: "Whole Slide Image Classification of Gastric Cancer using Convolutional Neural Networks"
_MIDL.amsterdam/2018/Conference — Submitted to MIDL 2018_

### Review · AnonReviewer2 · 2018-05-09
**This paper presents the gastric cancer detection system based on convolutional neural networks. The large-scale whole slide dataset has been introduced, and patch-level and slide-level classification pipeline has been proposed. Even if there is a weak technical novelty, they have shown the various experimental results.**

**Rating:** 3
**Confidence:** 3

**Review:**


Quality & Clarity

#1. This paper is well organized and clarified all components in the proposed system.
#2. The description of dataset and experimental results is well written.
#3. Authors have shown the impressive tools for visualization of heatmap results and annotation.

Originality & Significance

(+) They have introduced the large-scale whole slide image dataset.
(+) In order to find the appropriate patch size for cancer detection, they have performed experiments with various patch size and architectures. Also, the results have been well organized.
(+) They have proposed the complete pipeline including patch-level and slide-level prediction.
(-) Why do they train the network from scratch? It may be slightly better fine-tuning than scratch in terms of training time.
(-) In the slide-level prediction part, they have designed 9 features including various factors from cancer likelihood map. But, I don’t know what these features mean and why these features can improve the performance.
(-) They have shared the training dataset between patch-level and slide-level. Therefore, the slide-level prediction module based on random forest would be biased by patch-level training results.



**Special Issue:**

No

---

### Review · AnonReviewer1 · 2018-05-09
**Whole Slide Image Classification of Gastric Cancer using Convolutional Neural Networks**

**Rating:** 2
**Confidence:** 3

**Review:**

In this paper, whole-slide images of gastric cancer are classified for cancer detection using a pipeline based on convolutional networks.
Tumour regions are fist classified via patch classification, and a random forest classifier is trained to predict a label at image level, based on hand-crafted features extracted from the classification map.
Existing architectures, namely VGG-16, ResNet and DesneNet are used.
Internal dataset is used for training and validation.

It is not clear whether the training and the validation set were split in such a way that no intersection is present.

Equation 1 reports a formula for normalization of the values in the likelihood map, but it is not clear why this specific formula is applied.

The authors state that because of the difference between natural images (used to pre-train used networks) and histopathology images, fine tuning of models was not done and networks were trained from scratch.
Although this is a valid approach, specially because a large amount of patches can be extracted from histology images, it should be noted that it is well known that pre-trained networks can be fine-tuned to tackle image classification in medical imaging, and usually this does not hamper the optimization procedure.

Several performance scores reported in the paper have a value of 100%.
Although always desirable, it is often the effect of an imperfect data split (for example, intersection of training, validation and test data).

A validation on an external dataset would be needed, in order to show the robustness of the method when images stained and scanned in other labs are used, which is a well-known problem in HE-stained image classification in digital pathology.


**Special Issue:**

No

---

### Review · AnonReviewer3 · 2018-05-17

**Rating:** 2
**Confidence:** 3

**Review:**

The authors present a method for classification of whole-slide images as containing gastric cancer or not. The method itself is very straightforward and has been used extensively in challenges such as CAMELYON16. The comparison of the different architectures is a nice touch. The data set used is not publicly available, which is a shame. In general the paper is not clearly written. Especially the data set section is confusing and I had to reread it several times to understand what was going on. Still the information on for example training and validation splits is not mentioned. Results are generally good, but no external dataset was used, making it hard to assess the generalizability of the method. Given the fact that the method itself is not methodologically novel, I think this paper does not meet the criteria or acceptance.

**Special Issue:**

No

---

### Decision · Program_Chairs · 2018-05-15
**Paper32 Acceptance Decision**

Reject